# FCMpy: a python module for constructing and analyzing fuzzy cognitive maps

Samvel Mkhitaryan[1], Philippe Giabbanelli[2], Maciej K Wozniak[3], Gonzalo Nápoles[4], Nanne De Vries[1] and Rik Crutzen[1]

[1] Health Promotion, Maastricht University, Maastricht, Netherlands
[2] Computer Science & Software Engineering, Miami University of Ohio, Oxford, Ohio, United States
[3] Division of Robotics, Perception and Learning, KTH Royal Institute of Technology, Stockholm, Sweden
[4] Cognitive Sciences and Artificial Intelligence, Tilburg University, Tilburg, Netherlands

## ABSTRACT

FCMpy is an open-source Python module for building and analyzing Fuzzy Cognitive Maps (FCMs). The module provides tools for end-to-end projects involving FCMs. It is able to derive fuzzy causal weights from qualitative data or simulating the system behavior. Additionally, it includes machine learning algorithms (*e.g.*, Nonlinear Hebbian Learning, Active Hebbian Learning, Genetic Algorithms, and Deterministic Learning) to adjust the FCM causal weight matrix and to solve classification problems. Finally, users can easily implement scenario analysis by simulating hypothetical interventions (*i.e.*, analyzing what-if scenarios). FCMpy is the first open-source module that contains all the functionalities necessary for FCM oriented projects. This work aims to enable researchers from different areas, such as psychology, cognitive science, or engineering, to easily and efficiently develop and test their FCM models without the need for extensive programming knowledge.

## INTRODUCTION

Fuzzy Cognitive Maps (FCMs) were introduced by *Kosko (1986)* as an extension to the traditional cognitive maps and are used to model and analyze complex systems. FCMs are applied in a variety of fields such as engineering (*Stylios & Groumpos, 2004*), health sciences and medicine (*Papakostas et al., 2011*; *Salmeron & Papageorgiou, 2012*; *Giabbanelli, Torsney-Weir & Mago, 2012*), environmental sciences (*Kok, 2009*; *Papageorgiou & Kontogianni, 2012*), and political analysis (*Andreou, Mateou & Zombanakis, 2005*; *Giabbanelli, 2014*).

An FCM represents a system as a directed signed graph where components are represented as nodes and the causal relationships between these components are represented by weighted directed edges. The dynamics of the system are examined by simulating its behavior over discrete simulation steps. In general, FCMs can be constructed based on the inputs of domain experts (*i.e.*, expert-based FCMs), data collected about the system (*e.g.*, data-driven approaches) or the combination of the two (*i.e.*, hybrid approaches) (*Mkhitaryan et al., 2020*).



Corresponding author
Samvel Mkhitaryan,
s.mkhitaryan@
maastrichtuniversity.nl

**Table 1 FCMpy features and descriptions.**

| Feature (submodules) | Description |
| --- | --- |
| Expert FCM (fcmpy.ExpertFcm) | Methods for constructing expert based FCMs based on qualitative data. |
| Simulator (fcmpy.FcmSimulator) | Methods for running simulations on top of a defined FCM. |
| ML (fcmpy.NHL, fcmpy.AHL fcmpy.RCGA, fcmpy.ISE, fcmpy.OSE) | Learning algorithms for training FCMs and model evaluation. |
| Classification (fcmpy.ml.classification.ELTCN fcmpy.ml.classification.FCM_MP) | Classification algorithms based on FCMs. |
| Intervention (fcmpy.Intervention) | Methods for running scenario analysis. |

The available solutions for constructing and analyzing FCMs come in the form of dedicated software solutions (*e.g.*, *Mental Modeler*, *FCM Designer*), open source libraries (*e.g.*, *fcm* package available in R, *pfcm* available in Python) and open source scripts (*Firmansyah et al., 2019*; *Nápoles et al., 2018*). However, the available open source solutions provide only partial coverage of the useful tools for building and analyzing FCMs, lack generality for handling different use cases, or require modifying the source code to incorporate specific features (*Nápoles et al., 2018*). For example, the fcm and pfcm packages provide utilities for simulating FCMs but not for constructing them based on qualitative (*e.g.*, by applying fuzzy logic) or quantitative inputs. To our knowledge, none of the available open source solutions (*e.g.*, R and Python) implements learning algorithms for FCMs (*e.g.*, Non-Linear Hebbian Learning (NHL), Active Hebbian Learning (AHL), Real-coded genetic algorithm (RCGA)). Although several software packages have successfully implemented such algorithms (*e.g.*, FCM Expert, FCM Wizard), their reliance on a graphical user interface prevents their integration in a data science workflow articulated around a language such as Python (*Nápoles et al., 2018*).

The dedicated modules in our proposed FCMpy package provide utilities for (1) constructing FCMs based on qualitative input data (by applying fuzzy logic), (2) simulating the system behavior, (3) implementing learning algorithms (*e.g.*, Nonlinear Hebbian Learning, Active Hebbian Learning, Genetic Algorithms and Deterministic Learning) to optimize the FCM causal weight matrix and model classification problems, and (4) implementing scenario analysis by simulating hypothetical interventions (*i.e.*, analyzing what-if scenarios). Table 1 gives a compact listing of all major capabilities present in the FCMpy codebase.

## CONSTRUCTING EXPERT-BASED FCMS

Expert-based FCMs are often constructed based on data collected from the domain experts (*e.g.*, by the means of surveys) where the domain experts first identify the factors relevant to the problem domain and then express the causal relationships between these factors with linguistic terms (*e.g.*, very high, high, low). Fuzzy logic is subsequently applied to convert linguistic ratings into numerical weights (*i.e.*, crisp values). The conversion of linguistic ratings to numerical weights includes the following four steps: (1) define fuzzy membership functions for the linguistic terms, (2) apply fuzzy implication rule onto the

fuzzy membership functions based on the expert ratings, (3) combine the membership functions resulting from the second step with an aggregation operation, and (4) defuzzify the aggregated membership functions (*Mkhitaryan et al., 2020*; *Mago et al., 2012*, *2013*). In this section, we first describe methods for reading data from different file formats and then describe the methods for constructing expert-based FCMs based on qualitative data following the four steps described above.

## Data handling

The available open source solutions for expert-based FCMs do not provide utilities for working with different file types thus limiting their usability. Data on FCMs include the edges (represented as pairs of source/target) and the associated linguistic ratings of the survey participants. The `ExpertFcm` class provides a `read_data` method for reading data from .csv, .xlsx, and .json files (see the code snippet below).

The corresponding files should satisfy certain requirements that are described in detail in the PyPI documentation. Before using the `read_data` method we first need to define the linguistic terms (explained in detail in the next section) used in the data. We can do that by using the `linguistic_terms` method. The `read_data` requires the file path as an argument. The additional arguments that depend on the file extension (*e.g.*, csv, json, xlsx) should be specified as keyword arguments. For the .xlsx and .json files, when the optional `check_consistency` argument is set to `True` then the algorithm checks whether the experts rated the causal impact of the edges (source-target pairs) consistently in terms of the valence of the causal impact (positive or negative causality). If such inconsistencies are identified, the method outputs a separate .xlsx file that documents such inconsistencies.

```
>>> from fcmpy import ExpertFcm
>>> fcm = ExpertFcm()
>>> fcm.linguistic_terms = {
                            '-vh': [-1, -1, -0.75],
                            '-h': [-1, -0.75, -0.50],
                            '-m': [-0.75, -0.5, -0.25],
                            '-l': [-0.5, -0.25, 0],
                            '-vl': [-0.25, 0, 0],
                            'na': [-0.001, 0, 0.001],
                            '+vl': [0, 0, 0.25],
                            '+l': [0, 0.25, 0.50],
                            '+m': [0.25, 0.5, 0.75],
                            '+h': [0.5, 0.75, 1],
                            '+vh': [0.75, 1, 1]
                            }
>>> data = fcm.read_data (file_path = 'data_test.xlsx',
                          check_consistency = False,
engine = 'openpyxl')
```

The `read_data` method returns an ordered dictionary where the keys are the experts' IDs (or the names of the excel sheets in the case of an excel file or the row index in case of a csv file) and the values are `pandas` dataframes with the expert inputs.

It is often useful to check the extent to which the participants agree on their opinions with respect to the causal relationships between the edges. This is often done by calculating the information entropy (*Kosko, 1996*) expressed as:

$$R = -\sum_{i=1}^{n} p_i log_2(p_i) \tag{1}$$

where $p_i$ is the proportion of the answers (per linguistic term) about the causal relationship. The value of entropy is always greater than or equal to zero. If all the experts give the same answer about a particular edge (*e.g.*, from *C1* to *C3*), the entropy score for that connection will be 0. However, if experts disagree about the importance of a connection (*e.g.*, in the edge between *C1* and *C2*), the entropy value would increase. The entropy scores can be calculated with the `entropy` method (see the code snippet below).

```
>>> entropy = fcm.entropy (data)
From    To   Entropy
C1      C1   0.000000
        C2   1.459148
        C3   0.000000
        C4   0.000000
C2      C1   1.459148
        C2   0.000000
        C3   0.000000
        C4   0.000000
C3      C1   0.820802
        C2   0.000000
        C3   0.000000
        C4   0.930827
C4      C1   0.000000
        C2   0.000000
        C3   0.000000
        C4   0.000000
```

## FOUR STEPS FOR OBTAINING CAUSAL WEIGHTS

To convert the qualitative ratings of the domain experts to numerical weights *via* fuzzy logic, we must (1) define the fuzzy membership functions, (2) apply a fuzzy implication rule, (3) combine the membership functions, and (4) defuzzify the aggregated membership functions to derive the numerical causal weights. Table 2 gives a concise listing of the methods for building expert based FCMs.

**Table 2 FCMpy.ExpertFcm features and descriptions.**

| Feature | Description |
|---|---|
| Linguistic terms (fcmpy.ExpertFcm.linguistic_terms) | A setter for specifying the linguistic terms the experts used. |
| Universe of discourse (fcmpy.ExpertFcm.universe) | A setter for specifying the universe of discourse for the fuzzy membership functions. |
| Read data (fcmpy.ExpertFcm.read_data) | A method for reading data from different file formats. |
| Entropy (fcmpy.ExpertFcm.entropy) | A method for calculating entropy. |
| Generate membership functions (fcmpy.ExpertFcm.automf) | A method for automatically generating fuzzy membership functions. |
| Fuzzy implication (fcmpy.ExpertFcm.fuzzy_implication) | A method for applying fuzzy implication rules on the defined membership functions. |
| Fuzzy aggregation (fcmpy.ExpertFcm.aggregate) | A method for applying fuzzy aggregation rules on the activated membership functions. |
| Defuzzification (fcmpy.ExpertFcm.defuzz) | Defuzzification methods for calculating crisp value based on the aggregated membership functions. |
| Build (fcmpy.ExpertFcm.build) | A method for automatically building FCMs based on the qualitative input data. |

## Step 1: define fuzzy membership functions

Fuzzy membership functions are used to map the linguistic terms to a specified numerical interval (*i.e.*, universe of discourse). In FCMs, the universe of discourse is specified in the [−1, 1] interval where the negative causality is possible or in the [0, 1] interval otherwise. The universe of discourse can be specified with the `universe` setter (see the code snippet below).

```
>>> import numpy as np
>>> fcm.universe = np.arange(−1, 1.001,.001)
```

To generate the fuzzy membership functions we need to decide on the geometric shape that would best represent the linguistic terms. In many applications, a triangular membership function is used (*Zadeh, 1971*). The triangular membership function specifies the lower and the upper bounds of the triangle (*i.e.*, where the meaning of the given linguistic term is represented the least) and the center of the triangle (*i.e.*, where the meaning of the given linguistic term is fully expressed).

The `linguistic_terms` method sets the linguistic terms and the associated parameters for the triangular membership function (see the code snippet below).

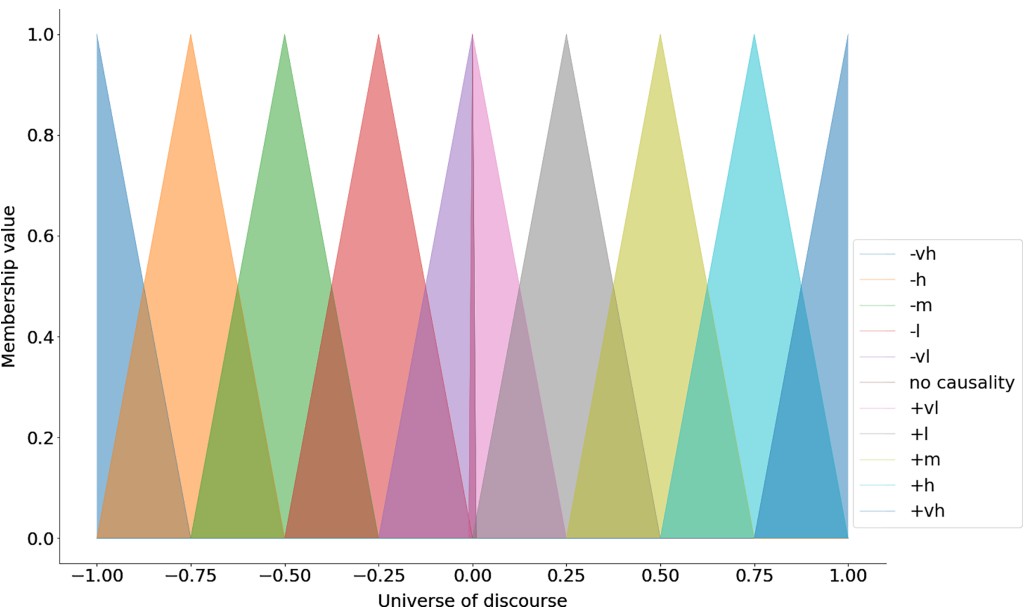

**Figure 1 Triangular membership functions.**

```
>>> fcm.linguistic_terms = {
                            '-VH': [-1, -1, -0.75],
                            '-H': [-1, -0.75, -0.50],
                            '-M': [-0.75, -0.5, -0.25],
                            '-L': [-0.5, -0.25, 0],
                            '-VL': [-0.25, 0, 0],
                            'No Causality': [-0.001, 0, 0.001],
                            '+VL': [0, 0, 0.25],
                            '+L': [0, 0.25, 0.50],
                              '+M': [0.25, 0.5, 0.75],
                              '+H': [0.5, 0.75, 1],
                              '+VH': [0.75, 1, 1]
                              }
```

The keys in the above dictionary represent the linguistic terms and the values are lists that contain the parameters for the triangular membership function (*i.e.*, the lower bound, the center and the upper bound) (see Fig. 1). After specifying the universe of discourse and the linguistic terms with their respective parameters one can use use the `automf` method to generate the membership functions (see the code snippet below).

```
>>> fcm.fuzzy_membership = fcm.automf(method = 'trimf')
```

In addition to the triangular membership functions, the `automf` method also implements gaussian membership functions ('gaussmf') and trapezoidal membership functions ('trapmf') (based on `sci-kit fuzzy` module in python).

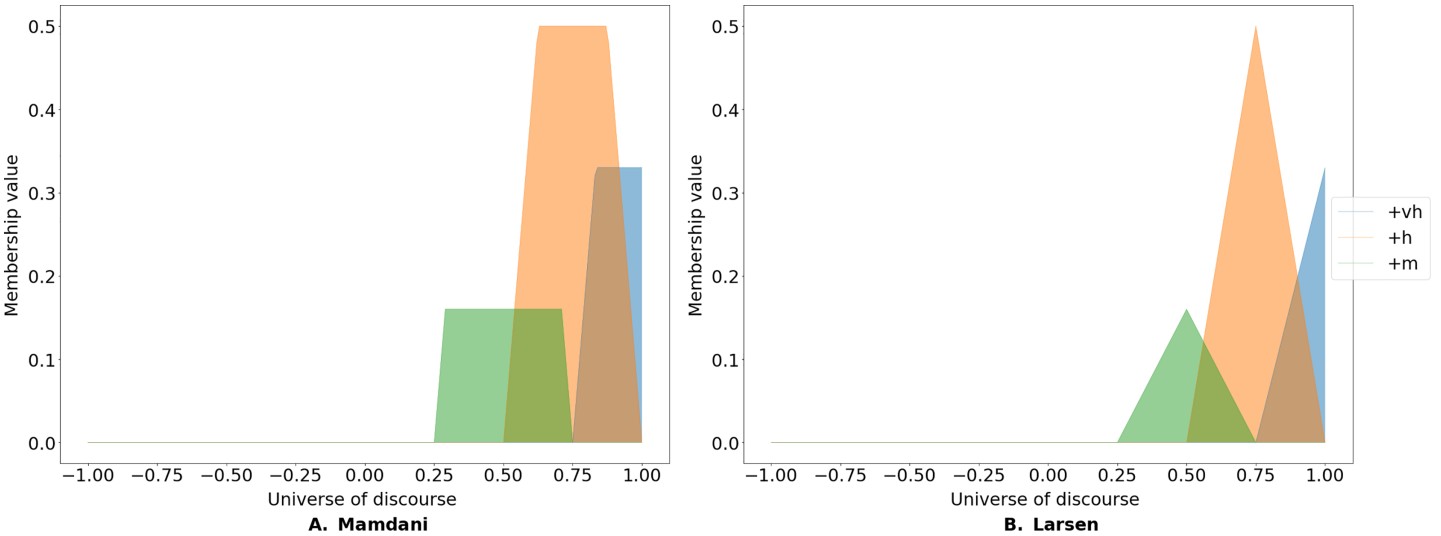

**Figure 2 Fuzzy implication rules.** FCMpy implements two types of fuzzy implication rules: the widely-used Mamdani minimum fuzzy implication (A) and Larsen's product fuzzy implication (B) rules.

**Table 3 Fuzzy implication rules.**

| Argument | Option | Description |
| --- | --- | --- |
| Method | 'Mamdani' | Mamdani's fuzzy implication rule |
| | 'Larsen' | Larsen's fuzzy implication rule |

## Step 2: apply the fuzzy implication rule

To determine the level of endorsement of the linguistic terms for a given pair of concepts, one must first identify the level of endorsement of the given terms by the participants. This is done by calculating the proportion of the answers to each linguistic term for a given edge. Consider a case where 50% of the participants (*e.g.*, domain experts) rated the causal impact of an antecedent on the consequent as Positive High, 33% rated it as Positive Very High and the 16% rated it as Positive Medium. Subsequently, a fuzzy implication rule is used to activate the corresponding membership functions. Two such rules are often used, namely Mamdani's minimum and Larsen's product implication rule (*Nandi, 2012*).

The Mamdani minimum fuzzy implication rule is expressed as:

$$\mu_R(x,y) = min(\mu_A(x), \mu_B(y)) \tag{2}$$

where $\mu_A(x)$ and $\mu_B(y)$ denote the membership value $x$ to the linguistic term $A$ and the membership value $y$ to the linguistic term $B$ respectively.

The Mamdani rule cuts the membership function at the level of endorsement (see Fig. 2A). In contrast, Larsen's implication rule re-scales the membership function based on the level of endorsement (see Fig. 2B) and is expressed as:

$$\mu_R(x,y) = \mu_A(x) \cdot \mu_B(y) \tag{3}$$

**Table 4 Aggregation rules.**

| Argument | Option | Description |
|---|---|---|
| Method | 'fMax' | Family maximum |
| | 'algSum' | Family Algebraic Sum |
| | 'eSum' | Family Einstein Sum |
| | 'hSum' | Family Hamacher Sum |

We can use `fuzzy_implication` method to apply the selected implication method (see the available methods in Table 3 and the code snippet below).

```
>>> mfs = fcm.fuzzy_membership
>>> act_pvh = fcm.fuzzy_implication(mfs['+vh'],
                        weight = 0.33, method = 'Mamdani')
>>> act_pm = fcm.fuzzy_implication(mfs['+m'],
                        weight = 0.16, method = 'Mamdani')
>>> act_ph = fcm.fuzzy_implication(mfs['+h'],
                        weight = 0.5, method = 'Mamdani')
>>> activatedMamdani = {'+vh': act_pvh,
                        '+h': act_ph, '+m': act_pm}
```

### Step 3: aggregate fuzzy membership functions

In the third step, we must aggregate the activated membership functions taken from the previous step. This is commonly done by applying the family maximum aggregation operation. Alternative methods for aggregating membership functions include the family Algebraic Sum (see Eq. (4)), the family Einstein Sum (see Eq. (5)) and the family Hamacher Sum (see Eq. (6)) (*Piegat, 2001*).

$$f(x,y) = x + y - x \cdot y \tag{4}$$

$$f(x,y) = \frac{(x+y)}{(1+x \cdot y)} \tag{5}$$

$$f(x,y) = \frac{(x+y-2 \cdot x \cdot y)}{(1-x \cdot y)} \tag{6}$$

One can use the `aggregate` method to aggregate the activated membership functions (see the available aggregation method in Table 4 and the code snippet below).

```
>>> import functools
>>> aggregated = functools.reduce(lambda x,y:
                fcm.aggregate(x=x, y=y, method='fMax'),
            [activatedMamdani[i] for i in activatedMamdani.
keys()])
```

where $x$ and $y$ are the membership values of the linguistic terms involved in the problem domain after the application of the implication rule presented in the previous step.

**Table 5 Defuzzification methods.**

| Argument | Option | Description |
|---|---|---|
| Method | 'centroid' | Centroid |
| | 'mom' | Mean of maximum |
| | 'som' | Min of max |
| | 'lom' | Max of maximum |

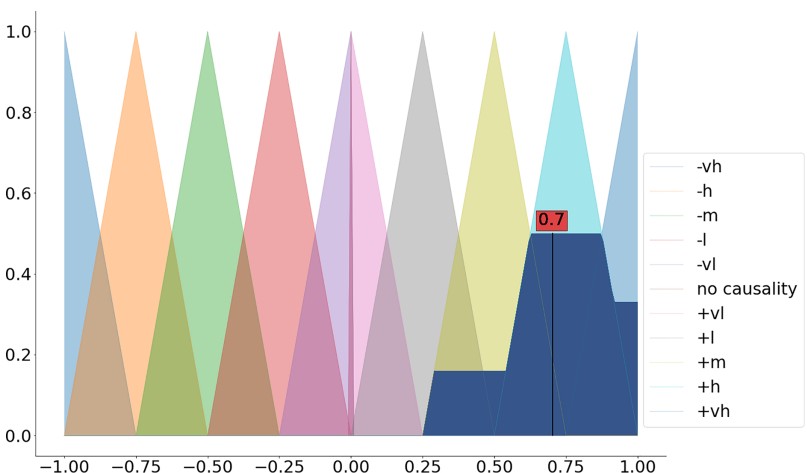

**Figure 3 Defuzzification of the aggregated membership functions.**

## Step 4: defuzzify the aggregated membership functions

The last step includes the calculation of the crisp value based on the aggregated membership functions (a.k.a. defuzzification). Among the available defuzzification methods (see the available defuzzification methods in Table 5) the most commonly used method is the centroid method (a.k.a. center of gravity) (*Stach et al., 2005*).

We can apply the dedicated `defuzz` method to derive the crisp value (see Fig. 3 and the code snippet below).

```
>>> dfuz = fcm.defuzz(x=fcm.universe,
        mfx=aggregated, method='centroid')
```

The above mentioned four steps can either be done and controlled independently as we have shown, or users can rely on a single `build` method that implements those steps to calculate the numerical weights for all the concept pairs in the data (see the code snippet below). The method returns a pandas dataframe with the calculated weights.

```
>>> weight_matrix = fcm.build(data=data,
            implication_method = 'Mamdani',
            aggregation_method = 'fMax',

            defuzz_method='centroid')
```

| Table 6 Defuzzification methods. | | |
|---|---|---|
| **Argument** | **Option** | **Description** |
| Inference | 'kosko' | Kosko |
| | 'mKosko' | Modified Kosko |
| | 'rescaled' | Rescaled |
| Transfer | 'sigmoid' | Sigmoid |
| | 'tanh' | Hyperbolic tangent |
| | 'bivalent' | Bivalent |
| | 'trivalent' | Trivalent |

```
          C1        C2        C3        C4
C1    0.000000  0.702032  0.000000  0.000000
C2    0.607453  0.000000  0.000000  0.000000
C3    0.555914  0.000000  0.000000  0.172993
C4    0.000000  0.000000  0.000000  0.000000
```

In this section, we showed how the structure of the FCMs can be derived based on data collected from the domain experts. In the subsequent section, we illustrate how the system behavior can be simulated on top of the defined FCM structure.

## SIMULATING THE SYSTEM BEHAVIOR WITH FCMS

The dynamics of the specified FCM are examined by simulating its behavior over discrete simulation steps. In each simulation step, the concept values are updated according to a defined inference method (*Papageorgiou, 2011b*). The `FcmSimulator` module implements the following three types of inference methods (see the available options in Table 6):

- Kosko:

$$A_i^{(t+1)} = f\left(\sum_{j=1}^{n} A_j^{(t)} \cdot w_{ji}\right) \tag{7}$$

- Modified Kosko:

$$A_i^{(t+1)} = f\left(A_i^{(t)} + \sum_{j=1}^{n} A_j^{(t)} \cdot w_{ji}\right) \tag{8}$$

- Rescaled:

$$A_i^{(t+1)} = f\left((2A_i^{(t)} - 1) + \sum_{j=1}^{n} (2A_j^{(t)} - 1) \cdot w_{ji}\right) \tag{9}$$

where $a_j^{(t)}$ is the value of concept $j$ at the simulation step $t$ and $w_{j,i}$ is the causal impact of concept $j$ on concept $i$. Note that a (transfer) function $f(x)$ is applied to the result. As shown in the equations above, this function is necessary to keep values within a certain range (*e.g.*,

[0,1] for sigmoid function or [−1,1] for hyperbolic tangent). In the current version, four such functions are implemented (see the available options in Table 6):

- Sigmoid:

$$f(x) = \frac{1}{1 + e^{-\lambda x}}, x \in \mathbb{R}; \text{binds node values to } [0, 1] \tag{10}$$

- Hyperbolic tangent:

$$f(x) = tanh(x) = \frac{sinh(x)}{cosh(x)} = \frac{e^{2x} - 1}{e^{2x} + 1}, x \in \mathbb{R}; \text{binds node values to } [-1, 1] \tag{11}$$

- Bivalent:

$$f(x) = \begin{cases} 1, & x > 0 \\ 0, & x \le 0 \end{cases}, x \in \mathbb{R}; \text{binds node values to } \{0, 1\} \tag{12}$$

- Trivalent:

$$f(x) = \begin{cases} 1 & x > 0 \\ 0 & x = 0 \\ -1 & x < 0 \end{cases}, x \in \mathbb{R}; \text{binds node values to } \{-1, 0, 1\} \tag{13}$$

where, $x$ is the value calculated by applying the above mentioned inference methods and the $\lambda$ is a steepness parameter for the sigmoid function.

The simulation is run until either of two conditions is met: (1) some concepts of interest have a difference lower than a given threshold between two consecutive steps (default value 0.001), or (2) a user-defined maximum number of iterations is reached. If we denote by S the activation vector for a subset of concepts of interest (*i.e.*, the outputs of the FCMs), then the first condition can be stated as:

$$\exists t \in 1, 2, \ldots, T - 1 : |S^{(t+1)} - S^{(t)}| < threshold \tag{14}$$

The `simulate` method takes the initial state vector and the FCM weight matrix (a.k.a., connection matrix) and applies one of the mentioned update functions over a number of simulation steps (see the simulation results using different inference and transfer methods in Figs. 4–6). One can specify the output concepts by supplying a list of these concepts to the respective `output_concepts` argument. If the `output_concepts` argument is not specified then all the concepts in the FCM are treated as output concepts and the simulation stops when all the concepts change by less than the threshold between two consecutive steps.

```
>>> import pandas as pd
>>> from fcmpy import FcmSimulator
```

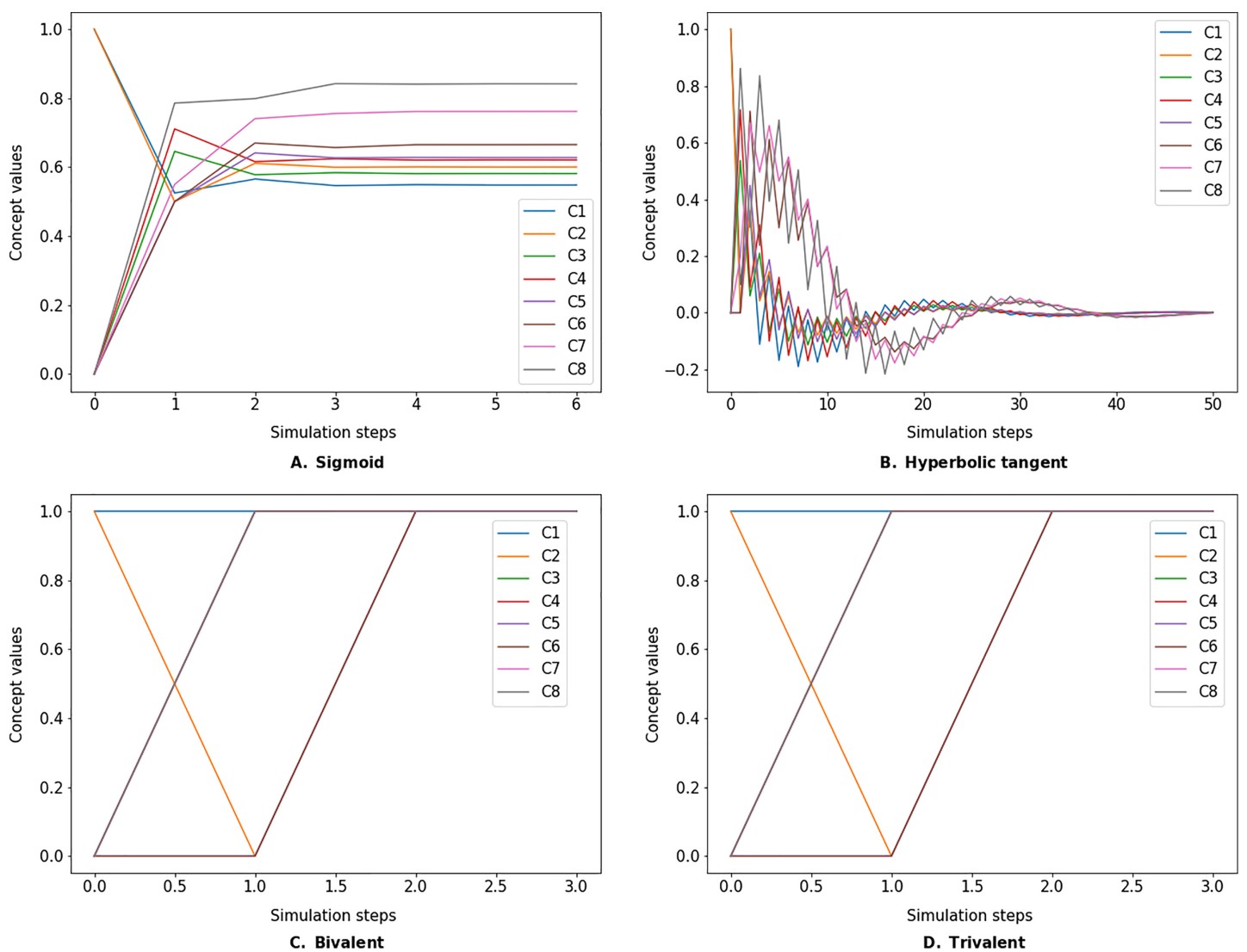

**Figure 4  FCM simulated with *Kosko* inference method and different transfer functions.** FCMpy supports Sigmoid (A), Hyperbolic tangent (B), Bivalent (C), and Trivalent (D) transfer methods.               

```
>>> w = np.asarray([[0.0, 0.0, 0.6, 0.9, 0.0, 0.0, 0.0, 0.8],
                    [0.1, 0.0, 0.0, 0.0, 0.0, 0.0, 0.2, 0.5],
                    [0.0, 0.7, 0.0, 0.0, 0.9, 0.0, 0.4, 0.1],
                    [0.4, 0.0, 0.0, 0.0, 0.0, 0.9, 0.0, 0.0],
                    [0.0, 0.0, 0.0, 0.0, 0.0, −0.9, 0.0, 0.3],
                    [−0.3, 0.0, 0.0, 0.0, 0.0, 0.0, 0.0, 0.0],
                    [0.0, 0.0, 0.0, 0.0, 0.0, 0.8, 0.4, 0.9],
                    [0.1, 0.0, 0.0, 0.0, 0.0, 0.1, 0.6, 0.0]])
```

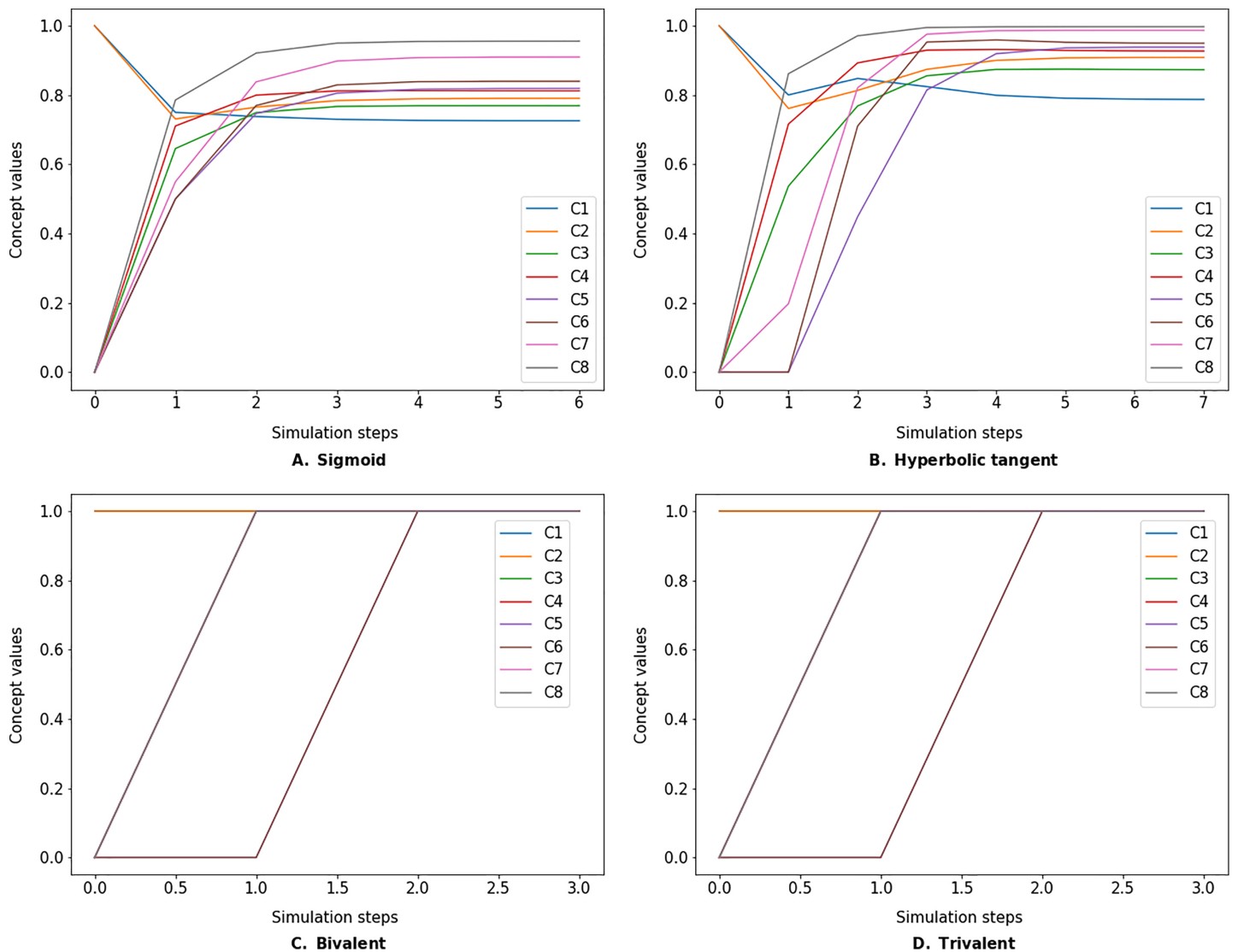

**Figure 5 FCM simulated with *modified Kosko* inference method and different transfer functions.** FCMpy supports Sigmoid (A), Hyperbolic tangent (B), Bivalent (C), and Trivalent (D) transfer methods.

```
>>> weight_matrix = pd.DataFrame(w,
                    columns=['C1','C2','C3','C4',
                    'C5','C6','C7','C8'])
>>> init_state = {'C1': 1, 'C2': 1, 'C3': 0,
                    'C4': 0, 'C5': 0,
                    "C6': 0, 'C7': 0, 'C8': 0}
>>> sim = FcmSimulator()
>>> res = sim.simulate(initial_state = init_state,
                    weight_matrix = weight_matrix,
                    transfer='sigmoid', inference='mKosko', l=1,
                    thresh = 0.001, iterations = 50)
```

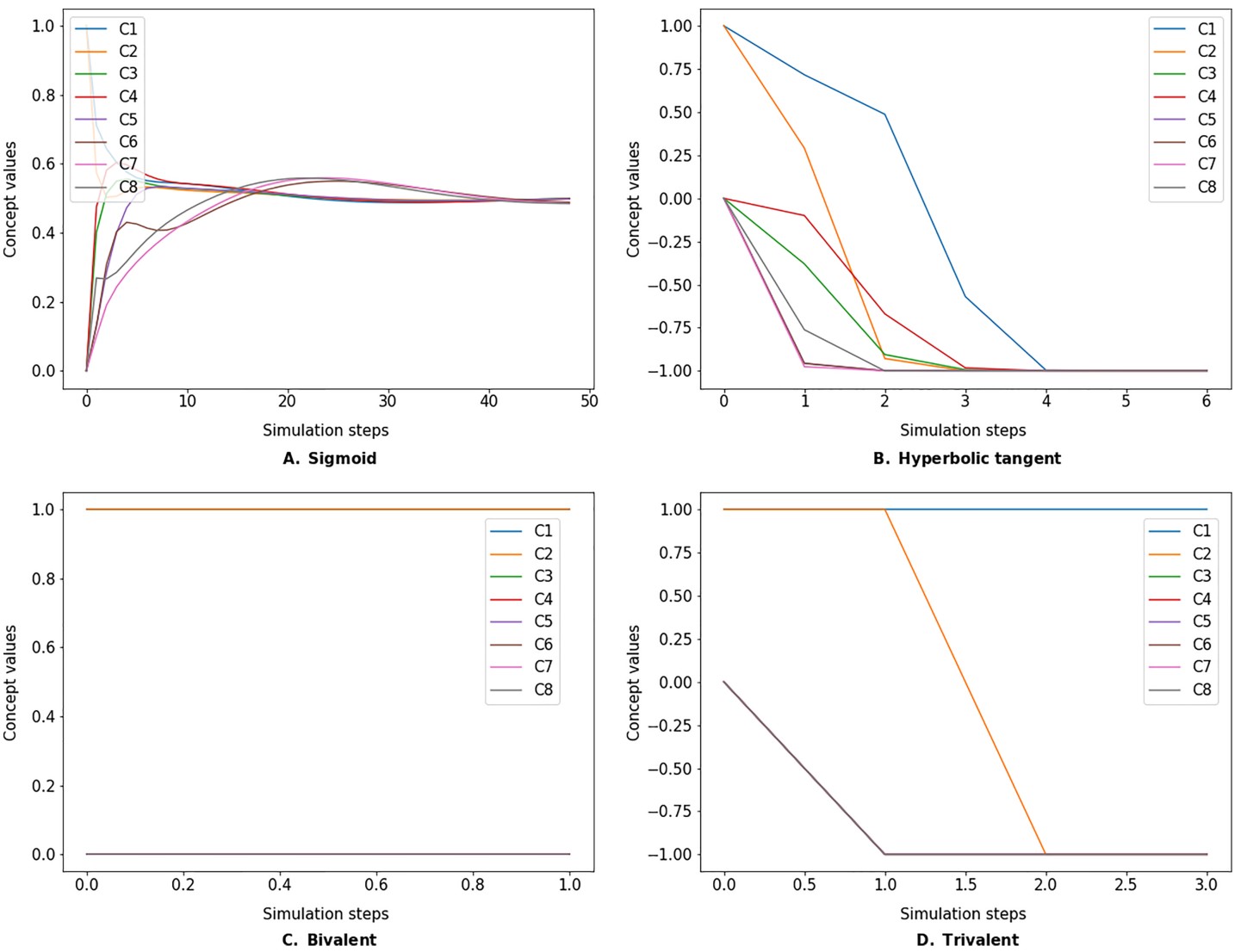

**Figure 6 FCM simulated with *rescaled* inference method and different transfer functions.** FCMpy supports Sigmoid (A), Hyperbolic tangent (B), Bivalent (C), and Trivalent (D) transfer methods.

```
The values converged in the 7 state (e <= 0.001)
          C1        C2   ...        C5        C6        C7        C8
0  1.000000  1.000000   ...  0.000000  0.000000  0.000000  0.000000
1  0.750260  0.731059   ...  0.500000  0.500000  0.549834  0.785835
2  0.738141  0.765490   ...  0.746700  0.769999  0.838315  0.921361
3  0.730236  0.784168   ...  0.805531  0.829309  0.898379  0.950172
4  0.727059  0.789378   ...  0.816974  0.838759  0.908173  0.954927
5  0.726125  0.790510   ...  0.818986  0.839860  0.909707  0.955666
6  0.725885  0.790706   ...  0.819294  0.839901  0.909940  0.955774
```

## LEARNING ALGORITHMS FOR FCMS

As shown in the previous sections, FCMs are often constructed based on experts' knowledge about the system. In certain domains of applications, modelers either optimize

the FCMs constructed by the experts and/or constructing FCMs entirely based on the data collected about the systems. A set of machine learning algorithms were developed to meet these tasks and have previously been applied to numerous fields such as the optimization of industrial processes (*Papageorgiou, Stylios & Groumpos, 2006*; *Stach, Kurgan & Pedrycz, 2008*; *Papageorgiou, 2011a*), decision making (*Poczeta, Papageorgiou & Gerogiannis, 2020*), and classification (*Nápoles et al., 2014*; *Nápoles, Jastrzębska & Salgueiro, 2021*). In the proposed library, we include three types of algorithms used for edge optimization, FCM generation, and classification. We used the state-of-the-art methods (*Nápoles, Jastrzębska & Salgueiro, 2021*; *Nápoles et al., 2020*) and the foundational ones that have been widely adopted (*Papageorgiou, Stylios & Groumpos, 2006*; *Stach, 2010*). In the following sections, we present examples of these algorithms. Additionally, we successfully tested some of these learning methods on a case study about nutrition which included 257 participants. We refer readers to our work for more information and for an illustration of the application of the presented library on a *real-world* use case (*Wozniak, Mkhitaryan & Giabbanelli, 2022*).

## Hebbian learning

One of the weaknesses of an FCM constructed by the experts is its potential convergence to undesired regions. For example, given an intervention scenario, the model may predict only extreme values such as 0 or 1 (*Lavin et al., 2018*). To overcome this weakness *Papageorgiou, Stylios & Groumpos (2006)* proposed two learning strategies, namely the Active Hebbian Learning (AHL) and the Non-Linear Hebbian Learning (NHL) algorithms that are based on the Hebbian learning rule (*Hebb, 2005*). The task of the proposed algorithms is to modify the initial FCM connection matrix constructed by the expert such that the chosen nodes (called Desired Output Concepts *DOCs*) always converge within the desired range. Both algorithms are similar to FCM simulation, with the main difference being that concepts' values and *weights* are updated at each time step, whereas during a simulation, only the concepts values are changing.

In the NHL algorithm, the activation values and weights are *simultaneously* updated at each time step. In AHL, nodes and weights are updated *asynchronously* based on a sequence of activation patterns specified by the user. During each simulation time step, a new node becomes an *"activated node"*; only this node and its incoming edges are updated, while everything else remains unchanged. Along with optimizing existing edges, AHL creates new connections between the concepts, which may be an undesirable behavior if the modeler's intent is to tweak the weights rather than create connections that have not been endorsed by experts.

The learning process continues until two termination conditions are fulfilled. First, the fitness function ($F_1$) is calculated for each DOC as shown in Eq. (15). If value of $F_1$ for each DOC declines at each time step, and the DOCs values are within a desired range, the first termination condition is fulfilled. Second, it is crucial to determine whether the values of the DOCs are stable, *i.e.*, if their values vary with each step more than a threshold $e$ shown in Eq. (16). This threshold should be determined experimentally, and it is recommended to

be set between 0.001 and 0.005 (*Papageorgiou, Stylios & Groumpos, 2006*). If the change is lower than the threshold, the second termination condition is fulfilled.

$$F_1 = \sqrt{|DOC_j^{(k)} - \frac{DOC_j^{min} - DOC_j^{max}}{2}|^2} \qquad (15)$$

$$F_2 = |DOC_j^{(k+1)} - DOC_j^{(k)}| < e \qquad (16)$$

If the termination conditions are satisfied then the learning process may stop, otherwise, it will continue until a maximum number of steps is reached (we set the default value to 100). In order to use these methods, the user has to provide the initial weight matrix, initial concept values, and the DOCs. In addition, these variables are necessary, in most cases, algorithms converge only for a specific combination of values of the hyperparameters: learning rate ($\eta$), decay coefficient ($\gamma$), and the slope of the sigmoid function. The sample values used in several case studies are slope [0.9, 1.01], decay for NHL [0.99, 1.0], decay for AHL [0.01, 0.1] and learning rate [0.001, 0.1]. The optimization of an FCM from a water tank case study (*Papageorgiou, Stylios & Groumpos, 2004*; *Ren, 2012*; *Papakostas et al., 2011*) using the algorithms above is demonstrated in the code snippet below.

```
>>> from fcmpy import NHL
>>> import numpy as np
# initial values of weight matrix
>>> w_init_WT = np.asarray([[0,-0.4,-0.25,0,0.3],
                            [0.36,0,0,0,0],
                            [0.45,0,0,0,0],
                            [-0.9,0,0,0,0],
                            [0,0.6,0,0.3,0]])
>>> w_init_WT = pd.DataFrame(w_init_WT,
                       columns=['C1', 'C2', 'C3', 'C4', 'C5'],
                       index = ['C1', 'C2', 'C3', 'C4', 'C5'])
# initial values of the concepts
>>> init_states_WT = {'C1': 0.40, 'C2': 0.7077,
                      'C3': 0.612, 'C4': 0.717, 'C5': 0.30}
# DOCs
>>> doc_values_WT = {'C1':[0.68,0.74], 'C5':[0.74,0.8]}
# NHL
>>> nhl = NHL(state_vector=init_states_WT,
          weight_matrix=w_init_WT, doc_values=doc_values_WT)
>>> res_nhl = nhl.run(learning_rate = 0.01, l=.98, iterations=100)
The NHL learning process converged at step 63 with the
learning rate eta = 0.01 and decay = 1!
        C1         C2         C3         C4         C5
C1  0.000000  -0.200310  -0.023806  0.000000  0.472687
C2  0.539068   0.000000   0.000000  0.000000  0.000000
C3  0.571531   0.000000   0.000000  0.000000  0.000000
C4 -0.832174   0.000000   0.000000  0.000000  0.000000
C5  0.000000   0.710523   0.000000  0.496934  0.000000
```

The `AHL.run` method has an additional `auto_learn` argument; if set to `True` then the algorithm automatically updates the hyperparameters during the learning process.

```
# AHL
>>> activation_pattern_WT = {0:['C1'], 1:['C2', 'C3'],
                                2: ['C5'], 3: ['C4']}
>>> ahl = AHL(state_vector=init_states_WT,
weight_matrix=w_init_WT,
                activation_pattern=activation_pattern_WT,
                doc_values=doc_values_WT)
>>> res_ahl = ahl.run(decay=0.03, learning_rate = 0.01, l=1,
                        iterations=100, transfer= 'sigmoid',
                        thresh = 0.002, auto_learn=False,
                        b1=0.003, lbd1=0.1, b2=0.005, lbd2=1)
The AHL learning process converged at step 19 with
the learning rate eta = 0.01 and decay = 0.03!
        C1        C2        C3        C4        C5
C1   0.000000 -0.128532 -0.060395  0.071200  0.218170
C2   0.245859  0.000000  0.068981  0.076592  0.074289
C3   0.288257  0.069457  0.000000  0.070342  0.068190
C4  -0.386349  0.073807  0.067187  0.000000  0.073991
C5   0.070113  0.368913  0.069145  0.223312  0.000000
```

If the learning process was successful (*i.e.*, the algorithm converged), the `run` method will return the optimized weight matrix as a dataframe. Successful outputs of NHL and AHL algorithms are presented in Fig. 7.

## Real-coded genetic algorithm (RCGA)

In certain domains of application, one has longitudinal data about the state variables included in the FCM and wants to find an FCM connection matrix that generates data that is close enough to the collected data (*Khan, Khor & Chong, 2004*; *Poczeta, Yastrebov & Papageorgiou, 2015*). In this regards, *Stach (2010)* proposed a real coded genetic algorithm for searching for an optimal FCM connection matrix. The proposed algorithm, named RCGA, builds on genetic algorithms where the search process includes the following six steps: (1) initialization, (2) evaluation, (3) selection, (4) recombination, (5) mutation, and (6) replacement.

In the initialization step, the algorithm generates a population of random solutions, that is a set of random weight matrices. Each solution is an $n \times n$ connection matrix. In the evaluation step, each candidate solution in the population is evaluated based on a fitness function shown in Eq. (17) and Table 7. We can observe how the fitness function is calculated on a simple example, shown in Fig. 8.

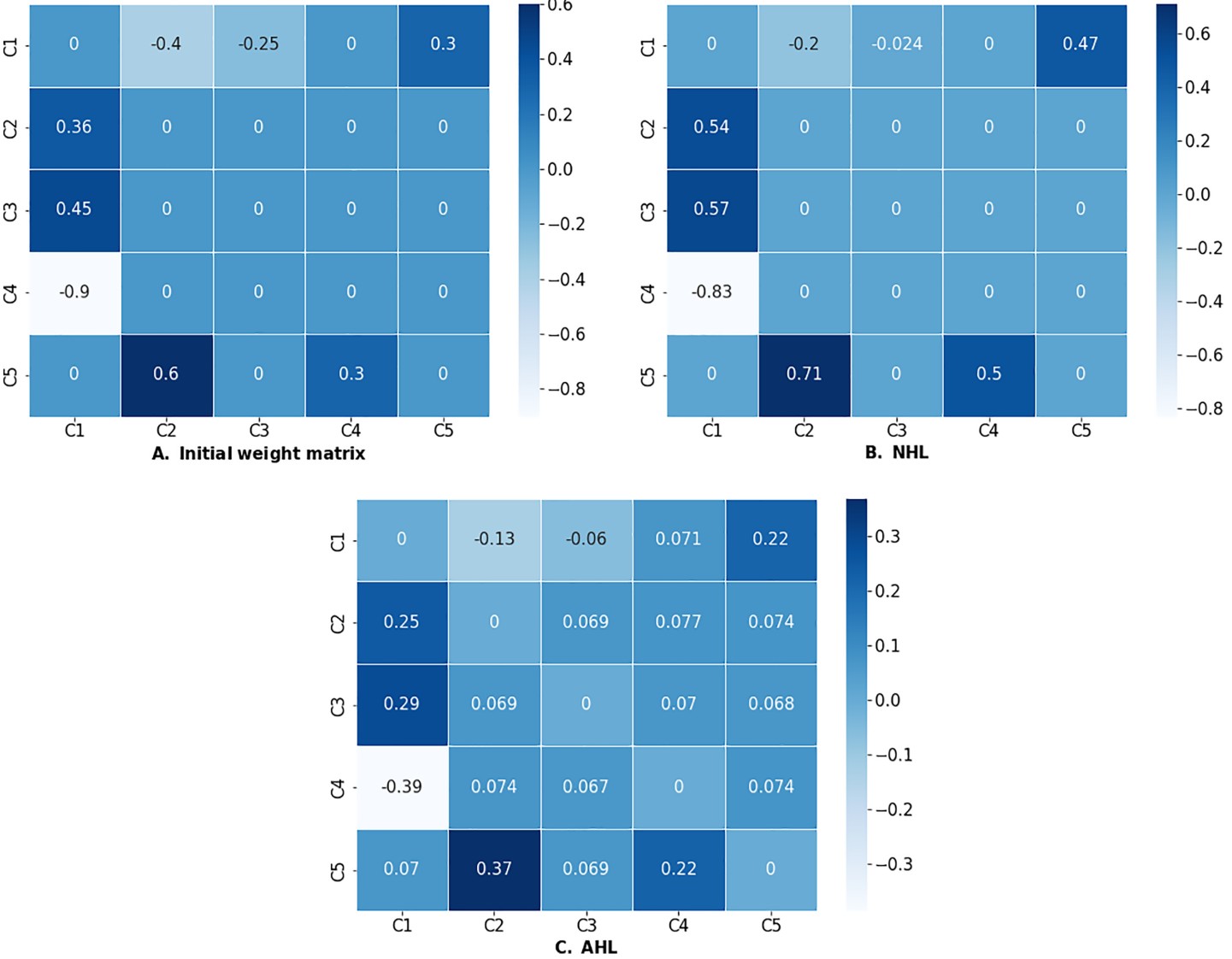

**Figure 7** (A) Initial weight matrix, (B) weight matrix optimized by NHL, and (C) weight matrix optimized by AHL.

**Table 7 Variables of fitness function used in RCGA algorithm.**

| Variable | Description |
| --- | --- |
| $\alpha$ | Normalization parameter |
| T | Number of chromosomes (elements in the generation) |
| N | Number of variables in the chromosome |
| C(t) | Observed data at time t |
| $\hat{C}(t)$ | State vector at step t |
| p | Defines the type of norm (default 1) |
| a | Defines the type of norm (default 100) |

**Figure 8  Example of fitness values calculation.**

**Table 8  Variables of fitness function used in RCGA algorithm.**

| Argument | Option | Description |
|---|---|---|
| Normalization_type | 'L1' | L1 normalization |
| | 'L2' | L2 normalization |
| | 'LInf' | L infinity normalization |
| Inference | 'kosko' | Kosko |
| | 'mKosko' | Modified kosko |
| | 'rescaled' | Rescaled inference method |
| Transfer | 'sigmoid' | Sigmoid |
| | 'bivalent' | Bivalent |
| | 'trivalent' | Trivalent |
| | 'tanh' | Hyperbolic tangent |

$$Error = \alpha \sum_{t=1}^{T-1} \sum_{j=1}^{N-1} |A_j(t) - \hat{A}_j(t)|^P$$
$$Fitness = \frac{1}{a * Error + 1}$$

(17)

In the selection step, candidate solutions are selected for mating (a.k.a., recombination). In each step, the algorithm randomly selects between two selection mechanisms (roulette wheel and tournament selection strategies). For the recombination step, the algorithm implements the recommended one point crossover operation with a probability of crossover specified by the user (`p_recombination`) (*Stach, 2010*). The crossover operation creates new solutions based on the solutions selected in the previous step. Next, the algorithm decides whether the new solutions produced in the previous step should undergo mutations. In the mutation step, the algorithm chooses between random and non-uniform mutation operations with a probability defined by the user (`p_mutation`). The replacement step is determined by the evolutionary approach specified by the user. The algorithm proposed by *Stach (2010)* is based on a generational approach where in each

step the new generation of solutions replace the old generation. Alternatively, the user could choose a steady state approach (a.k.a., SSGA) where in each step only two new solutions are produced and a decision is made whether the new chromosomes should be inserted back into the population. The current implementation of the SSGA uses a replacement strategy based on the concept of useful diversity (described in depth in *Lozano, Herrera & Cano (2008)*). To use the `RCGA` module, one needs to initialize the `RCGA` class by specifying the longitudinal data about the system, the population size, and the genetic approach to use (*i.e.*, generational or steady state). The additional parameters that can be modified by the user are presented in Table 8. Other parameters that can be modified by the user can be found in the documentation of the package available on PyPI.

The output of the learning process is the weight matrix with the highest fitness value throughout the search process. An example of generating FCM by the RCGA using historical data on water tank case study (*Papageorgiou, Stylios & Groumpos, 2004*) presented in the previous section is demonstrated in the code snippet below. We give the user an option to choose: `population_size` which is a number of weights matrices generated at each time step and `threshold` that is a minimum fitness value of at least one weight matrix in a generation, for the algorithm to succeed. To ensure the user does not get stuck in an infinite loop, we define `n_iterations` after which algorithm will terminate if the max fitness function of the *n_iterations* −1 generation was less than a `threshold`.

```
# Generate Longitudinal Data
>>> sim = FcmSimulator()
>>> data_WT = sim.simulate(initial_state=init_states_WT,
                           weight_matrix=w_init_WT, transfer='sigmoid',
                           inference='mKosko', thresh=0.001,
                           iterations=50, l=1)
# Select two time points
>>> data_WT = data_WT.iloc[:3]
# Generational Approach
>>> rcga = RCGA(data=data_WT, population_size=100,
                           ga_type='generational')
>>> rcga.run(n_iterations=30000, threshold=0.99)
# Steady State Approach
>>> rcga = RCGA(data=data_WT, population_size=100,
                           ga_type='ssga')
>>> rcga.run(n_iterations=30000, threshold=0.99)
```
The RCGA solution and the associated fitness score can be accessed in the `rcga.solution` and `rcga.fitness` fields.

```
>>> rcga.fitness
0.9790443073348711
>>> rcga.solution
          C1         C2         C3         C4         C5
C1    0.851948   0.550687   0.045434   0.648337   0.077349
C2   -0.870796  -0.944261  -0.684952  -0.829811   0.660863
C3    0.807396  -0.075391   0.223011   0.299976  -0.442272
C4   -0.479566   0.724956   0.048086   0.352487   0.120083
C5   -0.030473   0.068480   0.330551   0.003072  -0.347919
```

The learned FCM connection matrix can be validated by calculating the in-sample and out-sample errors by using the dedicated `ISE` and `OSE` modules (see the code snippet below) (*Stach, 2010*).

```
>>> from fcmpy import ISE
>>> from fcmpy import OSE
>>> val_ise = ISE()
>>> val_ose = OSE()
>>> error_ise = val_ise.validate(initial_state=init_states_WT,
                        weight_matrix=rcga.solution, data=data_WT,
                        transfer='sigmoid', inference='mKosko', l=1)
>>> error_ose, std = val_ose.validate(weight_matrix=rcga.
solution,
                        data=data_WT, low=0, high=1,
                        k_validation=100, transfer='sigmoid',
                        inference='mKosko', l=1)
```

## Classification algorithms

It is attractive for the researchers to choose FCMs for classification tasks, over other popular tools such as neural networks. This is because FCMs are easily explainable, which is a great advantage over *black box models* and, in many cases, equally accurate (*Nápoles, Jastrzębska & Salgueiro, 2021*; *Nápoles et al., 2020*). We give user a choice of two methods: Evolving Long-term Cognitive Networks (ELTCN) (*Nápoles, Jastrzębska & Salgueiro, 2021*) and deterministic learning (LTCN-MP)[1] (*Nápoles et al., 2020*).

LTCN-MP and ELTCN use the same topology (a fully connected FCM containing features nodes and class nodes) but the former produces numerical outputs (suitable for regression) while the latter produces nominal outputs (suitable for classification). In LTCN-MP and ELTCN algorithms (i) input variables are located in the inner layer and output variables in the outer layer, (ii) weights connecting the inputs are computed in an unsupervised way by solving a least squared problem, and (iii) weights connecting inputs with outputs are computed using the Moore-Penrose pseudo-inverse. Overall, we can say that LTCN-MP and ELTCN use the same topology but the former produces numerical outputs while the latter produces nominal outputs (decision classes). Additionally, the weights in the ELTCN model can change from one iteration to another.

An example of a model's structure with three features and three classes is shown in Fig. 9.

The user has to provide the path to the directory where the data file (*.arff* format) is located[2]. It is necessary that values of the features are normalized in the range between 0 and 1. Multiple data sets can be utilized and the results of each of them is saved in a dictionary, using the filename as a key. After running the learning process, the output consist of $k$ weight matrices, where $k$ is a number of validation folds (default 5) and the weight matrix of the connections between classes and feature nodes. We also provide users with automatically generated histograms showing values of the loss function and weights

[1] LSTCNs are a variant of FCMS where weights are not expected to be in the [−1,1] interval or have a causal meaning. We use regularization in order to keep them within that range.

[2] Currently, these two algorithms only accept *.arff* files, we are planning to accommodate more data files formats in the future versions.

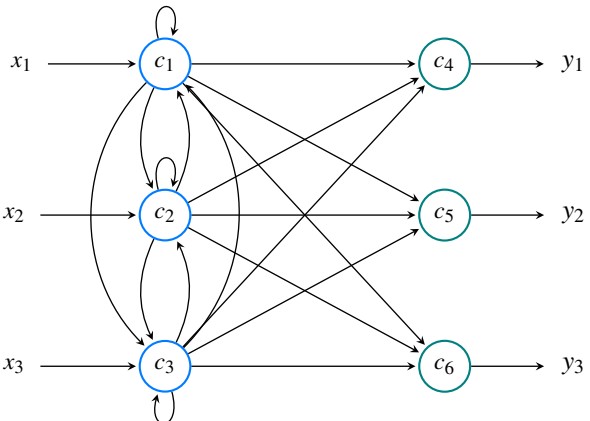

**Figure 9 Neural model comprised of $M = 3$ input neurons $(c_1, c_2, c_3)$ and $N = 3$ output neurons $(c_4, c_5, c_6)$.** Overall, the network has $P = M + N = 6$ neurons. In this example, the input signal is denoted as $x_i$ while output signal is denoted as $y_i$. These two values correspond to $a_i^{(0)}$ and $a_i^{(T)}$, respectively.

for each fold. In our examples we are using the *Iris data set*, a popular data set containing various measurements of iris flowers, such as sepal or petal length (*Fisher, 1936*).

```
>>> from fcmpy.ml.classification.eltcn import run
>>> path = 'data'
>>> results = run(path)

# average of the feature weight matrices
>>> print(results['irisnorm.arff']['avgW'])

# weight matrix connecting feature nodes with class nodes
>>> print(results['irisnorm.arff']['classW'])

[[ 0.4382333, −0.09812646, 0.79643613, 0.97163904],
[−0.2016794, 0.2925336, −0.21699643, 0.12452463],
[−0.0038293, −0.08575615, 0.47770625, 0.52030253],
[ 0.24917993, 0.21717176, 0.30113676, 0.34600133]]

[[−0.9135318, 0.15331155, 0.24873467],
[−1., 0.42972276, 0.17217618],
[−0.5610481, −0.58390266, 0.79575956],
[−0.60481995, −0.6718271, 0.72851056]]
```

LSTCN-MP focuses on discovering which and how features of the data set are important for the classification task as well as finding the weights connecting the inputs and outputs. Next, the LSTCN-MP algorithm outputs a 1-D array with values in the $[−1,1]$ range. The absolute values represent how important the features are for the classification task. In order to use the algorithm, the user has to provide the path of data sets as list under a key *sources* and then use that dictionary as an input to LSTCN-MP algorithm.

**Table 9 Keys and values of the input dictionary to the deterministic algorithm.**

| Keys | Values | Description | Default value |
|------|--------|-------------|---------------|
| 'M' | *int* | Output variables | 1 |
| 'T' | *int* | FCM Iterations | 1 |
| 'folds' | *int* | Number of folds in a (stratified) K-Fold | 10 |
| 'output' | *string* | Output csv file | './output.csv' |
| 'p' | *array* | parameters of *logit* and *expit* functions | {1.0, 1.0, 1.0, 1.0} |
| 'sources' | *array* | array with path of the dataset files | None |
| 'verbosity' | *bool* | verbosity | False |

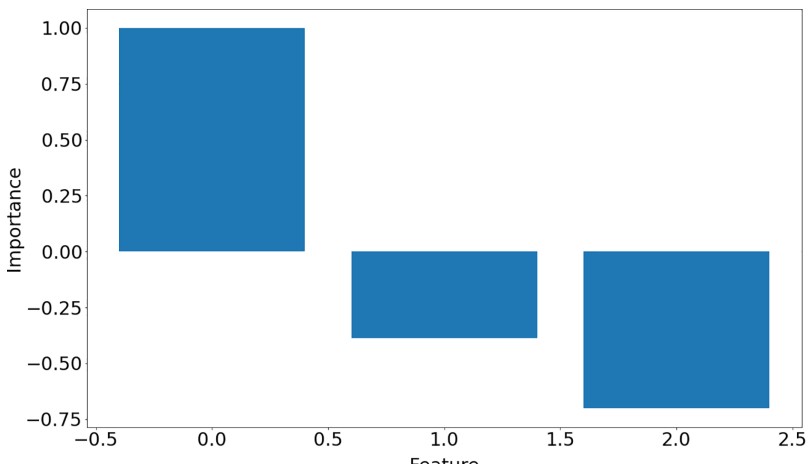

**Figure 10 Features importance for Iris data set.**

```
>>> sources = ['datasets/iris.arff','datasets/pima.arff']
>>> params = {'sources':sources}
```

Various hyperparameters can be used. Their default values and description can be found in *Nápoles et al. (2020)*, our library documentation, or Table 9.

```
>>> import fcmpy.ml.classification.FCM_MP as mp
>>> import matplotlib.pylab as plt
>>> sources = ['iris.arff']
>>> params = {'sources':sources}
>>> out = mp.run(**params)
# feature importance for classification purposes
>>> fig, ax = plt.subplots()
>>> ax.bar(range(len(out[0]['importance'].flatten())),
    height=out[0]['importance'].flatten())
# connections between features and class nodes
>> print(out['weights'])
[[ 0.46838854, −0.03855411, 1. ],
[−0.04176139, 0.46838854, −0.48478635],
[ 0.16611878, −0.07434725, 0.46838854]]
```

The function returns a list of dictionaries (one dictionary for each input data set) with keys representing hyperparameters used in the learning process, weight matrix, training error and importance of the features, which is shown in Fig. 10. Note that in Iris dataset, *feature* 0 is the most important feature whereas *feature* 1 is the least significant one in the decision-making.

In this section, we presented various implementations of learning algorithms that can be used to train FCMs based on expert inputs or quantitative data available about the system and solve classification problems. In the next section, we illustrate how FCMs can be used to analyze scenarios.

## SCENARIO ANALYSIS WITH FCMS

Scenario analysis in an FCM framework is often implemented by either changing the baseline values of the concepts (single shot interventions) or by introducing the proposed scenario as a new factor in the defined FCM and specifying the causal impact the proposed intervention has on the target concepts (continuous interventions). The single shot interventions mimic interventions that stop when a desired change in the specific target variables are achieved. In the continuous case, the intervention becomes part of the system and continuously impacts the target variables (*Giabbanelli & Crutzen, 2014*). The Intervention module provides the respective methods for analyzing different intervention cases. The module is instantiated by passing a simulator object to the constructor.

```
>>> from fcmpy import FcmSimulator, FcmIntervention
>>> inter = FcmIntervention(FcmSimulator)
```

Before specifying intervention cases and running simulations for each scenario, we need to create the baseline for the comparison (*i.e.*, run a simulation with baseline initial conditions and take the final state vector). To do this one needs to call initialize method. As in the FcmSimulator presented in the previous section, one can specify the output concepts by supplying a list of these concepts to the respective output_concepts argument. If the output_concepts argument is not specified then all the concepts in the FCM are treated as output concepts and the simulation stops when all the concepts change by less than the threshold between two consecutive steps.

```
>>> inter.initialize(initial_state=init_state,
                     weight_matrix=weight_matrix,
                     transfer='sigmoid', inference='mKosko',
                        thresh=0.001, iterations=50, l=1)
The values converged in the 7 state (e <= 0.001)
```

We can inspect the results of the initial simulation run (*i.e.*, 'baseline') in the test_results field as follows:

```
>>> inter.test_results['baseline']
```

```
          C1       C2    ...    C5       C6       C7       C8
0 1.000000 1.000000 ... 0.000000 0.000000 0.000000 0.000000
1 0.750260 0.731059 ... 0.500000 0.500000 0.549834 0.785835
2 0.738141 0.765490 ... 0.746700 0.769999 0.838315 0.921361
3 0.730236 0.784168 ... 0.805531 0.829309 0.898379 0.950172
4 0.727059 0.789378 ... 0.816974 0.838759 0.908173 0.954927
5 0.726125 0.790510 ... 0.818986 0.839860 0.909707 0.955666
6 0.725885 0.790706 ... 0.819294 0.839901 0.909940 0.955774
```

We can use the `add_intervention` method to specify the intervention cases. To specify a single shot intervention we must specify the name of the intervention and supply new initial states for the concept values as a dictionary (see the code snippet below).

```
>>> inter.add_intervention('intervention_1', type='single_shot',
                    initial_state = {'C1': 0.9, 'C2': 0.4})
```

For continuous intervention cases we must specify the name of the intervention, the concepts the intervention targets and the impact the intervention has on these concepts. In some cases we might be interested in checking scenarios where the intervention fails to be delivered to its fullest. For such cases we can specify the effectiveness of a given intervention case by setting the (optional) effectiveness argument to a number in the [0,1] interval (see the code snippet below). The effectiveness will decrease the *expected* causal strength of the intervention: for example, if an intervention is expected to reduce stress by 0.5 but is only 20% effective, then its actual reduction will be 0.1.

```
>>> inter.add_intervention('intervention_1', type='continuous',
                    impact={'C1':-.3, 'C2':.5}, effectiveness=1)
>>> inter.add_intervention('intervention_2', type='continuous',
                    impact={'C4':-.5}, effectiveness=1)
>>> inter.add_intervention('intervention_3', type='continuous',
                    impact={'C5':-1}, effectiveness=1)
```

In the example above, we specify three intervention cases. The first intervention targets concepts (nodes) C1 and C2. It negatively impacts concept C1 (−0.3) while positively impacting the concept C2 (0.5). We consider a case where the intervention has maximum effectiveness. The other two interventions follow the same logic but impact other nodes.

After specifying the proposed interventions, we can use the `test_intervention` method to test the effect of each case. The method requires the name of the intervention to be tested. Users also have the possibility of changing the number of iterations for the simulation; its default value is the same as specified in the initialization (see the code snippet below).

```
>>> inter.test_intervention('intervention_1', iterations=10)
>>> inter.test_intervention('intervention_2')
>>> inter.test_intervention('intervention_3')
The values converged in the 6 state (e <= 0.001)
```

```
The values converged in the 6 state (e <= 0.001)
The values converged in the 6 state (e <= 0.001)
```

The equilibrium states of the interventions can be inspected in the `equilibriums` field (see Fig. 11 and the code snippet below).

```
1>>> inter.equilibriums
      baseline  intervention_1  intervention_2  intervention_3
C1   0.725885   0.644651         0.715704         0.723417
C2   0.790706   0.870060         0.790580         0.790708
C3   0.769451   0.758786         0.768132         0.769141
C4   0.812473   0.798947         0.699316         0.812073
C5   0.819294   0.817735         0.819160         0.563879
C6   0.839901   0.838350         0.823430         0.871834
C7   0.909940   0.911004         0.909917         0.909778
C8   0.955774   0.954652         0.955427         0.952199
```

Lastly, one can inspect the differences between the interventions in relative terms (*i.e.*, % increase or decrease) compared to the baseline (see the code snippet below).

```
>>> inter.comparison_table
      baseline  intervention_1  intervention_2  intervention_3
C1   0.0       -11.191083       -1.402511        -0.339981
C2   0.0        10.035821       -0.015968         0.000202
C3   0.0        -1.385998       -0.171325        -0.040271
C4   0.0        -1.664794      -13.927524        -0.049314
C5   0.0        -0.190233       -0.016379       -31.175022
C6   0.0        -0.184640       -1.960979         3.802010
C7   0.0         0.116873       -0.002543        -0.017806
C8   0.0        -0.117365       -0.036331        -0.374038
```

## EXTENSIBILITY

Given the fact that new algorithms are continuously developed, the extensibility of the package is of paramount importance. Each module in the package uses a defined interface which ensures the scalability and cohesion of the package and their future extensions. Furthermore, we separated the creation of the objects from their use, to ensure that the future extensions do not cause major changes to the user code.

Let us examine how the package can be extended by considering a case where we want to add a new feature that allows us to read `.txt` file format. First, in the `reader.py` file we would need to define a new class called `TXT` which implements the `ReadData` interface. The interface has an abstract method called `read`.

Subsequently, we would need to add this object to the file called `methodsStore` which stores all the classes that support different file formats (*e.g.*, CSV, XLSX, JSON). The

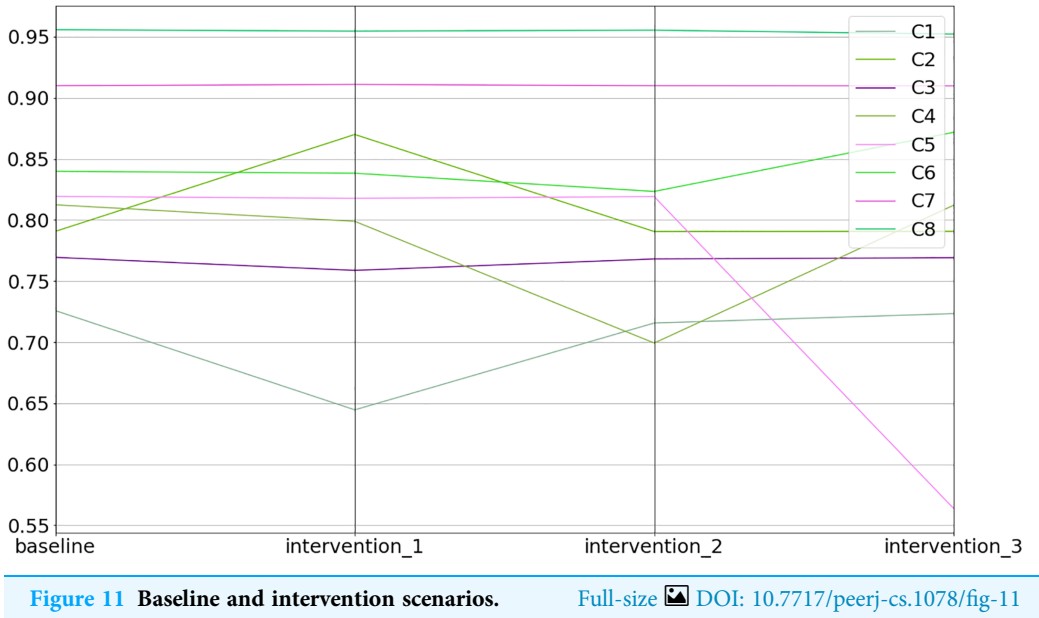

**Figure 11  Baseline and intervention scenarios.**

ReadStore class has a private attribute called `__methods` which is a dictionary that stores the argument we want the user to call in the `read_data` method as a key and the value is the Class we created for this particular file extension. The `get` method of the `ReaderStore` class takes the file extension as an argument and returns the corresponding object from the `__methods` dictionary. This `ReaderStore.get` method is called inside the `fcmpy.ExpertFcm.read_data` method, during which the appropriate file reader is supplied based on the user's specification.

## CONCLUSION

We hope that by providing researchers with such a tool we will promote using FCMs as one of the potential methods for different engineering tasks. We showed that FCMPy is a valuable tool for creating transparent and explainable behavioral models based on experts' answers. We provided detailed examples of how to create and inspect the model using different membership functions. Since one of the main purposes of FCMs is to monitor how the values of the concepts change throughout time, we provided the users with all the necessary options such as different inference methods and transfer functions. As a lot of research in the FCM field focuses on machine learning, we added several algorithms for weights optimization and data-driven model generation and made them easy to use. Finally, our library allows researchers to effortlessly examine how different interventions influence their model.

The FCMpy package provides a complete set of functions necessary to conduct projects involving FCMs. We created a tool that is open-source, easy to use, and provides the necessary functionality. The design and implementation of the tool results from a collaboration with multiple experts from the field of FCMs. We believe that this tool will facilitate research and encourage new students and scientists to involve FCMs in their projects. We included both well-known algorithms as well as recently developed ones. We

are planning to constantly update our library and welcome all scientific community contributions.

### Funding
The authors received no funding for this work.

### Competing Interests
The authors declare that they have no competing interests.

### Author Contributions
- Samvel Mkhitaryan conceived and designed the experiments, performed the experiments, analyzed the data, performed the computation work, prepared figures and/ or tables, authored or reviewed drafts of the article, and approved the final draft.
- Philippe Giabbanelli conceived and designed the experiments, performed the experiments, analyzed the data, performed the computation work, authored or reviewed drafts of the article, and approved the final draft.
- Maciej Wozniak conceived and designed the experiments, performed the experiments, analyzed the data, performed the computation work, prepared figures and/or tables, authored or reviewed drafts of the article, and approved the final draft.
- Gonzalo Nápoles conceived and designed the experiments, performed the experiments, performed the computation work, prepared figures and/or tables, authored or reviewed drafts of the article, and approved the final draft.
- Nanne De Vries conceived and designed the experiments, authored or reviewed drafts of the article, and approved the final draft.
- Rik Crutzen conceived and designed the experiments, authored or reviewed drafts of the article, and approved the final draft.

### Data Availability
The replication script for the examples presented in the article and the data files for running the replication scripts are available in the Supplemental Files.

### Supplemental Information
Supplemental information for this article can be found online at http://dx.doi.org/10.7717/ peerj-cs.1078#supplemental-information.

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
