# Peer review of "FCMpy: a python module for constructing and analyzing fuzzy cognitive maps"

_PeerJ Computer Science, doi:10.7717/peerj-cs.1078_

## Round 0.1 · original submission · Major Revisions

We have received two reports for the paper. In addition to the comments raised, the paper can be improved by considering the following points: 1. the abstract can be restructured without the lists but focusing more on the novelty and difficulty of the proposed method. 2. The information entropy expression can be better explained with some appropriate background. 3. More comparison study is needed for the section SIMULATING THE SYSTEM BEHAVIOR WITH FCMS.

·

Basic reporting

The Abstract need to be improved. I suggest that the author improve it.
Fig 5b was supposed to be NHL see line 355-360 while Fig 5c should be AHL see line 378-382
The English Language used is okay but can still be improved on.

Experimental design

This design meet the standard

Validity of the findings

The validity of the finding meet standard

Additional comments

I comment the authors for the good work done and the set data used. In addition, the manuscript is clearly written in clear, professional unambiguous language. If there is a weakness, it is in the abstract which should be improved upon before acceptance.

Reviewer 2 ·

Basic reporting

The language of the article is clear. The diagrams used are clear. The different sections and sub-sections of the paper need to be numbered. The structure of the paper could be improved further so that the flow of information from one section to another to be better.

Experimental design

The research question is well-defined and it is relevant to the scope of the journal. The methods applied are described with the necessary detail. There is detailed information on the different methods applied, however it would be interesting if the authors selected a specific, real-life problem (eg. a large complex decision making problem) and show the impact of FCMpy on the implementation of Machine Algorithms algorithms for FCMs. This is only a suggestion though as the authors do this already up to a specific degree.

Validity of the findings

The section of Conclusions is very concise and it needs to be enhanced.

Additional comments

Overall, an interesting paper. The authors need to enhance the conclusions of the paper as this is an important section of the paper.

---

## Round 0.2 · accepted · Accept

The reviewers' comments have been addressed. I recommend it for publication.

·

Basic reporting

No comment.

Experimental design

No comment

Validity of the findings

No comment

Additional comments

All suggestions and comments already updated in the new manuscript